Assessing the reproducibility of discriminant function analyses

Andrew Rose L. 1 2
Albert Arianne Y.K. 3
Renaut Sebastien 2 4
Rennison Diana J. 2
Bock Dan G. 2
Vines Tim 2 5 vines@zoology.ubc.ca
1 School of Environmental and Rural Science, University of New England , Armidale, NSW , Australia
2 Biodiversity Research Centre, University of British Columbia , Vancouver, BC , Canada
3 Women’s Health Research Institute, BC Women’s Hospital and Health Centre , Vancouver, BC , Canada
4 Institut de recherche en biologie végétale, Département de sciences biologiques, Université de Montréal , Montreal, QC , Canada
5 Molecular Ecology Editorial Office , Vancouver, BC , Canada
Wilke Claus
Electronic publication date: 2015 Aug 4
Publication date: 2015
Volume: 3
Electronic Location ID: e1137
Received 2015 Feb 13; Accepted 2015 Jul 8
Copyright: © 2015 Andrew et al.
Copyright year: 2015
Copyright holder: Andrew et al.
License: This is an open access article distributed under the terms of the Creative Commons Attribution License, which permits unrestricted use, distribution, and reproduction in any medium, provided the original author and source are credited.
License URL: https://creativecommons.org/licenses/by/3.0/

Keywords: Data curation, Repeatability, Data archiving, Statistics

Funding: NSERC postgraduate studentship NSERC Vanier CGS Killam doctoral scholarship NSERC postdoctoral fellowship Diana J. Rennison was supported by an NSERC postgraduate studentship. Dan G. Bock was supported by an NSERC Vanier CGS and a Killam doctoral scholarship. Sebastien Renaut was supported by an NSERC postdoctoral fellowship. The funders had no role in study design, data collection and analysis, decision to publish, or preparation of the manuscript.

==============================
Data are the foundation of empirical research, yet all too often the datasets underlying published papers are unavailable, incorrect, or poorly curated. This is a serious issue, because future researchers are then unable to validate published results or reuse data to explore new ideas and hypotheses. Even if data files are securely stored and accessible, they must also be accompanied by accurate labels and identifiers. To assess how often problems with metadata or data curation affect the reproducibility of published results, we attempted to reproduce Discriminant Function Analyses (DFAs) from the field of organismal biology. DFA is a commonly used statistical analysis that has changed little since its inception almost eight decades ago, and therefore provides an opportunity to test reproducibility among datasets of varying ages. Out of 100 papers we initially surveyed, fourteen were excluded because they did not present the common types of quantitative result from their DFA or gave insufficient details of their DFA. Of the remaining 86 datasets, there were 15 cases for which we were unable to confidently relate the dataset we received to the one used in the published analysis. The reasons ranged from incomprehensible or absent variable labels, the DFA being performed on an unspecified subset of the data, or the dataset we received being incomplete. We focused on reproducing three common summary statistics from DFAs: the percent variance explained, the percentage correctly assigned and the largest discriminant function coefficient. The reproducibility of the first two was fairly high (20 of 26, and 44 of 60 datasets, respectively), whereas our success rate with the discriminant function coefficients was lower (15 of 26 datasets). When considering all three summary statistics, we were able to completely reproduce 46 (65%) of 71 datasets. While our results show that a majority of studies are reproducible, they highlight the fact that many studies still are not the carefully curated research that the scientific community and public expects.

Introduction

Published literature is the foundation for future research, so it is important that the results reported in scientific papers be supported by the underlying data. After all, we cannot easily predict which aspects of a paper will prove useful in the future (Wolkovich, Regetz & O’Connor, 2012), and if a portion of the results are wrong or misleading then subsequent research effort may well be wasted (e.g., Begley & Ellis, 2012). One relatively simple way to judge the validity of published research is to obtain the original data analyzed in the paper and attempt to repeat some or all of the analyses: this allows researchers to retrace the path the authors took between the raw data and their results. Reproducibility in research is of growing interest and has recently gained traction with journals (Announcement: Reducing our irreproducibility, 2013; McNutt, 2014). There is clearly a need to quantify the validity of published research, yet there have been only a modest number of published studies that have tried to reproduce the results of published papers (e.g., Ioannidis et al., 2009; Gilbert et al., 2012; Errington et al., 2014), most likely because it is often difficult to access the underlying data (Wicherts et al., 2006; Savage & Vickers, 2009; Drew et al., 2013; Vines et al., 2013).

Even when the data file is available, one common problem that hampers reanalysis is poor data curation: it is sometimes difficult to relate the dataset provided by the authors upon request or archived at publication to the one described in the paper (Michener et al., 1997; Ioannidis et al., 2009; Gilbert et al., 2012). For example, variable names may differ between the obtained dataset and the one described in the study, or there may be differences in the number of variables or data points (see White et al., 2013). It is typically not possible to reproduce the authors’ analyses in these cases, and moreover the data may not be considered sufficiently reliable for testing new hypotheses.

The current study had two goals: to assess (a) how often poor data curation prevented re-analysis, and (b) how often we could reproduce authors’ results when the obtained dataset did match the one described in the paper. We made use of 100 datasets acquired from authors as part of an earlier study assessing the effect of time since publication on data availability (Vines et al., 2014). The articles we chose had to (i) contain morphometric data from plants or animals, (ii) have analysed the morphometric data with a Discriminant Function Analysis (DFA), and (iii) have not previously made the data available online. To make our study manageable in size, we selected only articles published in odd years (between 1991 and 2011), as detailed in Vines et al. (2014).

We focused on morphometric data because it has been collected in a similar fashion for decades (e.g., with Vernier callipers or a binocular microscope), so datasets from a range of time periods are expected to be similar in size and format. We used similar logic when selecting the analysis to reproduce: Discriminant Function Analysis (Fisher, 1936) has been applied to morphometric datasets for many decades. The function gives the best linear combination of morphometric variables that distinguishes between two or more known groups (e.g., sexes, populations, species). Typically, a sample of individuals with known affiliations is used to find the minimum set of variables that distinguishes the groups, and a discriminant function composed of the chosen variables is then used to classify unknown individuals. While computer processing power has greatly increased over the years, the way the analysis has been performed has remained the same. We can therefore reasonably compare DFAs from papers with a wide range of publication dates, allowing us to investigate how changing analysis software or date of publication affect reproducibility. In combination with (Vines et al., 2014), our results quantify the extent of the challenges facing science publication, both in terms of acquiring the original data analysed in the paper, and in terms of the proportion of analyses that are poorly curated or cannot be reproduced.

Materials and methods

As part of the Vines et al. (2014) study, we received 100 datasets from authors. For papers reporting a classical DFA of morphometric data, linear or quadratic DFA were considered, as were stepwise analyses where (a) the variables in the final model were presented and (b) at least one of three common metrics (see below) was presented. This allowed us to attempt to reproduce the final model as a simple linear DFA. Studies employing stepwise analysis of relative warps or Fourier-transformed data were also excluded at this point, as these studies unfortunately did not indicate which variables were included in the final model. A study entirely written in a foreign language (Spanish) was also excluded.

For each remaining study, we followed the protocol below.

(1) We first assessed the description of the methodology, checking whether the paper adequately described the groupings and morphometric variables used in the analysis.

(2) We examined the data files (in some cases multiple files were supplied), which sometimes required specialised file formats to be converted to text. This was carried out using the R packages ‘foreign’ (R Core Team, 2014) and ‘RODBC’ (Ripley & Lapsley, 2013). If the data file was clearly wrong (e.g., a summary table, instead of raw data) we categorized the paper as ‘Incorrect data file.’

(3) We assessed whether the metadata contained in the data file, in other files supplied by the author or in the accompanying email were complete and could be related to their description in the paper. We classified papers missing sample or variable names and those with unclear population groupings as having ‘Insufficient metadata.’ This category also included papers for which variable labels were in a foreign language and, even when translated in English (using http://translate.google.ca), could not be matched to the variables reported in the paper. However, we accepted files with unlabeled data columns where the identity of each column could be verified using information in the paper or supplied by the authors; the latter information came either from the files provided or their email message.

(4) We then went through the data deleting rows that contained missing data or other samples that the paper stated were not included in the analysis. Once this was completed, we identified discrepancies between the paper and the dataset in terms of sample sizes or number of variables, and categorized papers for which variables were missing or for which sample sizes did not match those reported in the paper as ‘Data discrepancy.’

(5) In addition to simple transformations (logarithm or square root), we conducted size adjustments based on multigroup principal components analysis (e.g., Burnaby’s (1966) back-projection) using the R packages ‘multigroup’ (Eslami et al., 2014) and ‘cpcbp’ (Bolker & Phillips, 2012).

(6) When there was more than one DFA analysis that met our criteria in a paper, we selected only the first one. We recorded whether raw or standardised coefficients were presented, whether cross-validation was used in the classification of individuals, and the statistical software used. The year of publication was recorded for each paper.

Based on a preliminary survey of the papers, we identified three DFA metrics to reproduce: the percentage of variance explained (PVE), the percentage of samples assigned correctly (PAC), and the largest model coefficient. These three summary statistics are commonly reported for DFAs, and are useful for interpreting DFA in a meaningful manner (Reyment, Blackith & Campbell, 1984), although the detail in which DFAs are described varies greatly depending on the focus of the paper. PVE and PAC are complementary indicators of the discriminatory power of a discriminant function, whereas the function coefficients make up the formula for assigning unknown samples to one group or another.

Our reanalysis procedure was designed to produce a single value for each metric per paper. Where possible, we compared the PVE for the first axis, which explains the greatest amount of variance in the model. When PVE was reported as the sum of the first two or three axes, we compared the summed PVE. We calculated the overall PAC, or the PAC for a particular group if the overall percent assigned correctly was not reported in the paper. For the coefficient, we selected the variable with the largest absolute coefficient, and determined from the paper whether the raw or standardised coefficient was used.

Although the original analyses used diverse statistical packages, we performed all discriminant function reanalyses in the statistical software R v3.2.0 (R Core Development Team, 2011), using the function ‘lda’ (in the MASS package; Venables & Ripley, 2002) with default parameters. Within R, we estimated each summary statistic using proportional or flat priors and used the value that was closest to the published value. Authors reported a variety of methods for assignment when calculating PAC, ranging from standard classification functions based on all data, to omitting one quarter of the data as a validation set. In our reanalysis, classification was carried out using leave-one-out (jackknife) cross-validation or direct prediction in ‘lda’, based on the description of the analysis in the paper. When authors did not specify whether cross-validation or direct prediction was used, we performed both and selected the value that was closest to the published result. While this approach biases the results towards the published value, it avoids unfair treatment of studies that used default parameters for their chosen software.

Our R code is provided in the Supplemental Information 1, along with example datasets from Gugerli, 1997, Berzins, Gilchrist & Burness (2009a), Berzins, Gilchrist & Burness (2009b) and Dechaume-Moncharmont, Monceau & Cezilly (2011). We considered the analysis to have been reproduced if the PVE, coefficient, or PAC was within 1% of the published value (termed ‘matched’), or was ‘close’ if it was within 5% of the published value.

We used generalised linear models (the core ‘glm’ function in R) to assess whether publication year affected the likelihood of problems in the data sets that would prevent attempts to reproduce the DFA results. Given a binomial model, we tested the effect of publication year on categorizing a study as ‘Insufficient metadata’ or ‘Data discrepancy’ (see points 3–4 above for category descriptions). A Fisher’s exact test was used to test the effect of statistical software on data problems and on the success of the reanalysis. We combined software used in just one study (S-Plus, STATGRAPHICS, and LINDA (Cavalcanti, 1999)) into a single category (“other”).

Although we contacted authors again to ask for their preferences regarding acknowledgment or anonymity, we did not seek further information (e.g., metadata or analysis parameters) to inform our reanalysis. We also note that we are unfortunately unable to provide all 100 of the datasets reanalysed here, in an attempt to make this study reproducible: the email sent by Vines et al. (2014) requesting these datasets stated ”Our analysis will not identify individual studies, but will instead focus on overall patterns and trends. Your responses will therefore be completely confidential.” Obtaining additional permission to make all datasets public alongside this paper would be very challenging, as evidenced by the 66% response rate to our email merely asking authors how they would prefer their work to be cited.

Results

The current study used 100 data sets originally gathered by Vines et al. (2014). Fourteen of those data sets were excluded from our reanalysis attempt (Tables 1 and 2): one paper was entirely in a language other than English (Spanish); two did not perform classical DFA; two used non-morphological data in their DFA; six did not present any of the metrics that we were attempting to reproduce; and three were based on stepwise analysis for which the final set of Fourier-transformed variables or relative warps were not specified.

Table 1 Summary of papers excluded from or included in the study, in total and listed by the statistical software originally used to analyse the data.

Those included in the study are further broken down by the reasons that reanalysis was not attempted or by the results of the reanalysis. The reanalysis outcome was classified as a complete match when all reanalyzed summary statistics were within 1% of the published values, a partial match when at least one (but not all) met this criterion, and no match when none met this criterion. The metrics considered were PVE, a discriminant function coefficient, and PAC.

Software	Excluded	Included	Incorrect data file	Insufficient metadata	Data discrepancy	No match	Reanalysed partial match	Complete match	
TOTAL	14	86	2 (2.3%)	7 (8.1%)	7 (8.1%)	12 (14%)	46 (53.5%)	12 (14%)	
JMP	2	2	0 (0%)	1 (50%)	0 (0%)	0 (0%)	0 (0%)	1 (50%)	
MATLAB	1	2	0 (0%)	0 (0%)	1 (50%)	0 (0%)	0 (0%)	1 (50%)	
R	0	5	2 (40%)	0 (0%)	1 (20%)	1 (20%)	0 (0%)	1 (20%)	
SAS	1	15	0 (0%)	3 (20%)	2 (13%)	3 (20%)	2 (13%)	5 (33%)	
SPSS	6	30	0 (0%)	0 (0%)	2 (7%)	5 (17%)	6 (20%)	17 (57%)	
STATISTICA	0	9	0 (0%)	1 (11%)	1 (11%)	2 (22%)	0 (0%)	5 (56%)	
SYSTAT	0	8	0 (0%)	0 (0%)	0 (0%)	1 (12%)	2 (25%)	5 (62%)	
Other	1	2	0 (0%)	1 (50%)	0 (0%)	0 (0%)	0 (0%)	1 (50%)	
Unknown	3	13	0 (0%)	1 (8%)	0 (0%)	0 (0%)	2 (15%)	10 (77%)	

Table 2 Published results and reanalyzed values of DFAs based on data files received from authors.

DFAs included in the current study were categorized according to the adequacy of data files and metadata, and the reproducibility of three metrics (percent variance explained, the largest coefficient and percent assigned correctly) among those that were able to be reanalyzed. Category indicates whether the data set was excluded from the study (E), was incorrect (I), had inadequate metadata (M), displayed data discrepancies (D) or was reanalysed (R). The reasons for excluding data sets from the study or preventing us from reanalyzing the data are summarized. The reanalysis outcome was classified as a complete match (C) when all reanalyzed summary statistics were within 1% of the published values, a partial match (P) when at least one (but not all) met this criterion, and no match (N) when none met this criterion. The same classification was applied to studies using the ‘close’ criterion (within 5%).

Study no.	Year	Software	PVE	COEF	PAC	Categ.	Reason	Reanalysis outcome	Citationa	
			Published	Reanalyzed	Published	Reanalyzed	Published	Reanalyzed			Match (within 1%)	Close (within 5%)		
1	1991	SAS	47.3	45.8			93.2	93.2	R		P	C	(Semple, Chmielewski & Leeder, 1991)	
2	1993	SAS	83.2	84.2	18.94	20.609			R		N	P	(Heraty & Woolley, 1993)	
3	1995	Other (STATGRAPHICS)	79.1	79.1	2.87	−2.868	72	71.9	R		C	C	(Darbyshire & Cayouette, 1995)	
4	1995	SPSS			0.892	0.7	100	100	R		P	P	(Cadrin, 1995)	
5	1995	SPSS	57.3	57.3			91.4	91.4	R		C	C		
6	1995	SPSS			4.02	−3.805	100	100	R		P	P	(Ruedi, 1995)	
7	1995	SYSTAT			−1.09	1.091	92	86.9	R		P	P		
8	1995	SYSTAT			2.115	−2.115	100	100	R		C	C	(Floate & Whitham, 1995)	
9	1997	Not stated							E	Not all variables are morphological			(Vanclay, Gillison & Keenan, 1997)	
10	1997	SPSS	67	66.9					R		C	C	(Brysting, Elven & Nordal, 1997)	
11	1997	SPSS	96.7	92.6	1.5	−2.488	100	98.6	R		N	P	(Gordo & Bandera, 1997)	
12	1997	SYSTAT	99.5	99	−0.57	0.611	89	88.7	R		P	P	(Gugerli, 1997)	
13	1999	Not stated							M	Row groupings don’t match paper				
14	1999	Not stated							E	No PVE, coef. or PAC				
15	1999	SAS	65	64.2			61	61.4	R		P	C		
16	1999	SPSS							E	No PVE, coef. or PAC				
17	1999	SPSS					73.4	73.4	R		C	C		
18	1999	SYSTAT					90	91.7	R		N	C		
19	2001	Not stated					100	100	R		C	C	(Rigby & Font, 2001)	
20	2001	SAS	96.7	96.4					R		C	C		
21	2001	SAS					71.3	93.8	R		N	N		
22	2001	SPSS			−1.072	−1.072	96	100	R		P	C	(Palma et al., 2001)	
23	2001	SPSS					100	100	R		C	C		
24	2001	SPSS	96	96					R		C	C	(Fernández & Feliner, 2001)	
25	2001	SPSS			5.228	−5.228	86	82.6	R		P	C	(Katoh & Tokimura, 2001)	
26	2001	STATISTICA					94.4	94.4	R		C	C		
27	2003	Not stated					90.3	90.3	R		C	C	(Okuda, Ito & Iwao, 2003)	
28	2003	Not stated			−2.176	−2.176	90.6	90.6	R		C	C		
29	2003	SAS							M	Column labels in Spanish				
30	2003	SAS							D	Extra rows				
31	2003	SPSS			1.011	1.011	100	100	R		C	C		
32	2003	SPSS			3.5		81		D	Extra rows			(Mills & Côté, 2003)	
33	2003	SPSS							D	Missing rows and row assignments unclear				
34	2003	SPSS					88.9	87.5	R		N	C		
35	2003	SPSS			0.772	0.766	84.3	84.3	R		C	C		
36	2003	STATISTICA							M	Column labels unclear				
37	2003	SYSTAT			1.28	−1.275	81	80.6	R		C	C	(Wicht et al., 2003)	
38	2005	JMP							E	No PVE, coef or PAC			(Nishida, Naiki & Nishida, 2005)	
39	2005	Not stated					79.9	79.7	R		C	C	(Hendriks, Van Duren & Herman, 2005)	
40	2005	Not stated	83	83.1			73	74.3	R		P	C		
41	2005	Not stated					100	100	R		C	C	(Radloff et al., 2005)	
42	2005	Other (S-Plus)							E	No PVE, coef or PAC				
43	2005	Other (LINDA)							M	Unclear groups			(Contrafatto, 2005)	
44	2005	SAS							M	Column labels missing			(Zaitoun, Tabbaa & Bdour, 2005)	
45	2005	SAS					94.3	94.9	R		C	C	(Marhold et al., 2005)	
46	2005	SPSS					46	38.2	R		N	N	(Aparicio et al., 2005)	
47	2005	SPSS	55.1	55.6	0.352	0.779	71.8	70.3	R		P	P		
48	2005	STATISTICA	67.5	67					R		C	C		
49	2005	STATISTICA					97	98.8	R		N	C		
50	2005	SYSTAT					100	100	R		C	C		
51	2007	MATLAB							D	Missing columns and insufficient metadata				
52	2007	Not stated			1.1	1.097	97	96.6	R		C	C	(Svagelj & Quintana, 2007)	
53	2007	Not stated					87.9	87.9	R		C	C	(De la Hera, Pérez-Tris & Telleria, 2007)	
54	2007	SAS			8.623	3.495	97.3	98.6	R		N	P		
55	2007	SAS					76	76.6	R		C	C	(Williams, Dean Kildaw & Loren Buck, 2007)	
56	2007	SAS							D	Missing columns			(Pearce, Fields & Kurita, 2007)	
57	2007	SPSS							E	No PVE, coef or PAC				
58	2007	SPSS					76.9	76.9	R		C	C	(Rioux-Paquette & Lapointe, 2007)	
59	2007	SPSS			0.689	0.647	100	85.4	R		N	N	(Santiago-Alarcon & Parker, 2007)	
60	2007	SPSS	61.8	61.6					R		C	C		
61	2007	SPSS							E	Final model not given			(Conde-Padín, Grahame & Rolán-Alvarez, 2007)	
62	2007	SPSS					84	83.3	R		C	C		
63	2007	STATISTICA					96.1	96.2	R		C	C		
64	2007	STATISTICA	93.3	93.3	−0.951	−0.951	89.2	89.2	R		C	C		
65	2007	STATISTICA			1.68	1.678	83.7	83.7	R		C	C	(Bourgeois et al., 2007)	
66	2007	SYSTAT	90.4	90.4			90	90	R		C	C		
67	2009	Not stated					91.2	91.2	R		C	C		
68	2009	Not stated							E	Not DFA				
69	2009	Not stated	40.8	41.1			79	78.3	R		C	C	(Hermida et al., 2009)	
70	2009	Not stated			0.242	0.084	100	100	R		P	P	(Buczkó, Wojtal & Jahn, 2009)	
71	2009	SAS	69	69.2	1.05	−1.053			R		C	C	(Pérez-Farrera et al., 2009)	
72	2009	SAS			0.95	0.604	80	80	R		P	P		
73	2009	SPSS					100	100	R		C	C		
74	2009	SPSS							E	Data not morphological				
75	2009	SPSS					76.4	77	R		C	C	(Thorogood, Brunton & Castro, 2009)	
76	2009	STATISTICA							D	Missing rows				
77	2009	STATISTICA					100	98.1	R		N	C		
78	2009	SYSTAT			2.8	2.795	91	91.5	R		C	C	(Berzins, Gilchrist & Burness, 2009a)	
79	2011	JMP							E	No PVE, coef or PAC			(Hata et al., 2011)	
80	2011	JMP							M	Column labels unclear				
81	2011	JMP			−7.06	7.063	100	100	R		C	C	(Gabrielson, Miller & Martone, 2011)	
82	2011	MATLAB	65.5	65					R		C	C	(Salcedo et al., 2011)	
83	2011	MATLAB							E	Not classical DFA			(Capoccioni et al., 2011)	
84	2011	Not stated	90	90.5					R		C	C	(Russell et al., 2011)	
85	2011	R							D	Missing rows				
86	2011	R							I	Wrong file				
87	2011	R							I	Wrong file				
88	2011	R	58	88.3			56	57.1	R		N	P		
89	2011	R					80.4	80.4	R		C	C	(Dechaume-Moncharmont, Monceau & Cezilly, 2011)	
90	2011	SAS							E	Spanish				
91	2011	SAS					100	100	R		C	C	(Parent, Plourde & Turgeon, 2011)	
92	2011	SPSS	81.8	81.7					R		C	C	(Forster, Ladd & Bonser, 2010)	
93	2011	SPSS	97.7	97.7			87.5	87.5	R		C	C	(Amado et al., 2011)	
94	2011	SPSS	58.3	58.3			62.9	62.9	R		C	C	(Ibáñez & O’Higgins, 2011)	
95	2011	SPSS	87.7	87.5					R		C	C		
96	2011	SPSS							E	Final model not given				
97	2011	SPSS							E	Final model not given			(Asanidze, Akhalkatsi & Gvritishvili, 2011)	
98	2011	SPSS					100	100	R		C	C		
99	2011	SPSS					95.7	93.9	R		N	C		
100	2011	SPSS	96	89.7	1.202	0.068	100	100	R		P	P		
Notes.

a Authors were contacted individually once reanalyses were performed. Only authors wishing to be identified are cited above. In addition, several authors agreed to be cited, but not identified directly (Amini, Zamini & Ahmadi, 2007; Audisio et al., 2001; Bulgarella et al., 2007; Ekrt et al., 2009; Foggi, Rossi & Signorini, 1999; Ginoris et al., 2007; Gouws, Stewart & Reavell, 2001; López-González et al., 2001; Magud, Stanisavljević & Petanović, 2007; Malenke, Johnson & Clayton, 2009; Schagerl & Kerschbaumer, 2009; Wasowicz & Rostanski, 2009).

Of the 86 remaining studies, the data files provided for two (2.3%) were classified as ‘Incorrect data file’: summary tables were provided instead of morphometric data, or the data set provided was used for a different analysis from the same paper (Fig. 1, Table 1). Six others (7.0%) were categorized as ‘Insufficient metadata’, as columns in the data files could not be matched to the variables described in the paper. Assignment as ‘Insufficient metadata’ was due to a combination of incomprehensible abbreviations in column headings (studies #36 and #80 in Table 2), missing column labels (#44), unclear row groupings (#13 and #43) and the use of a language other than English (study #29 in Table 2). Seven were classified as ‘Data discrepancy’: five data sets (5.8%) did not match the expected sample sizes, and two (2.3%) were missing variables.

Figure 1 Summary of the reproducibility of the 71 reanalyzed data sets and of the problems preventing reanalysis of 15 papers (see Table 1).

We found no effect of publication year on the probability of having ‘Insufficient metadata’ (odds ratio 0.98 (95%CI [0.84–1.15]), P = 0.82) and no effect of year on the probability of incorrect or inconsistent data (‘Data discrepancy’ or ‘Incorrect data’: odds ratio 1.06 (95%CI [0.90–1.26]), P = 0.45). Combining these main types of data problems that prevented us from attempting reanalysis (‘Incorrect data file,’ ‘Insufficient metadata,’ ‘Data discrepancy’), there was no effect of year (odds ratio 1.06 (95%CI [0.95–1.20]), P = 0.28).

Where stated, the type of software (SAS® (SAS Institute, Cary, North Carolina, USA), SYSTAT (SYSTAT Software Inc., Richmond, California, USA), SPSS (SPSS Inc., Chicago, Illinois, USA), MATLAB (Mathworks, Natick, Massachusetts, USA), STATISTICA (Statsoft, Tulsa, Oklahoma, USA), JMP (SAS Institute, Cary, North Carolina, USA), R (R Core Development Team, 2011), S-Plus® (TIBCO Software Inc., Palo Alto, California, USA), STATGRAPHICS (StatPoint Inc., Rockville, Maryland, USA) and LINDA (Cavalcanti, 1999)) used for the initial study had a significant effect on the probability of data problems (Fisher’s exact test, P = 0.012). This was largely due to a high likelihood of data problems (26.7%) among data sets originally analysed with SAS, compared with the overall proportion (17.4%; see Table 1).

We attempted a reanalysis of the DFA for the remaining 71 studies, and the results are summarised in Table 2. Our results regarding the PVE were generally close to the published values (Pearson correlation coefficient, r = 0.94, P < 0.0001; Fig. 2). Of the 26 reanalysed data sets reporting this statistic, our reproduced value was within 1% of the published value in 20 cases (80%), and within 5% of the published value in 24 cases (92%). The PAC statistic was also often reproduced (Pearson’s r = 0.95, P < 0.0001; Fig. 3). Of 60 reanalyses of PAC attempted, our values differed from the published value by 1% or less in 44 (73%) cases, while 55 (93%) were within 5%. Discriminant function coefficients were reproduced less frequently in the reanalysis. Using the absolute value of the coefficient to exclude sign differences, reproduced values were within 5% of the published value for 15 (58%) of the 26 data sets reanalysed for this statistic, and each of these values was also within 1%. There was still a strong correlation between the published value and our estimate (using absolute values, Pearson’s r = 0.96, P < 0.0001; Fig. 4).

Figure 2 PVE values from reanalysis versus published DFA. Points on the 1:1 line represent analyses differing by 1% or less.

Figure 3 PAC values from reanalysis versus published DFA. Points on the 1:1 line represent analyses differing by 1% or less.

Figure 4 Discriminant function coefficients from the reanalysis versus the published results.

Absolute values are used because the signs of coefficients depends on the order of variables. Points on the 1:1 line represent analyses differing by 1% or less.

Of all 112 reanalysed PVE, PAC and coefficient values, 79 (71%) were within 1% of the published value, and 95 (85%) were within 5% (Table 2). Considering the reported summary statistics together for each paper, our reanalysis failed to replicate any value in the paper at the most stringent level (within 1%) in 12 studies (17% of the total 71 data sets; Table 1); however, we were able to partially reproduce 13 (18%) studies and completely reproduce the results in 46 studies (65%). The reanalysed values were within 5% of the published value for all three statistics for 56 (79%) of studies.

There was no effect of publication year on discrepancies between the published and our values for PVE, coefficients or PAC (Fisher exact test, P > 0.2 in each case). Sample sizes were sufficient for a reliable test of the software effect for PAC only and this effect was not significant (Fisher’s exact test, P = 0.81). There was also no effect of software on the overall reanalysis success (Fisher’s exact test, P = 0.81).

Discussion

Confidence in scientific research is boosted when published results can be independently reproduced by other scientists (Price, 2011). Assuming that the raw data can be obtained (which is typically difficult, e.g., Wicherts et al., 2006; Wicherts, Bakker & Molenaar, 2011; Vines et al., 2013; Vines et al., 2014), several obstacles still remain. First, poor data curation (e.g., unintelligible column headings or missing samples) or inadequate methods description can mean that the dataset obtained cannot be matched to the one described in the paper, preventing reanalysis at the outset. Second, even when the datasets do match, some aspects of the results may be inherently harder to reproduce than others, perhaps because there are multiple calculation methods for the same summary statistic, or because the calculation involves ‘random walk’ estimation (e.g., Gilbert et al., 2012). By following recommended data sharing practices (e.g., White et al., 2013), we can aim towards reproducible research, and re-usable datasets.

In this paper we attempted to reproduce the results of Discriminant Function Analyses (DFA) for 100 datasets from papers published between 1991 and 2011. In contrast to the striking decline in data availability over time (Vines et al., 2014), we found no evidence that the reproducibility of DFAs decreased with time since publication. There was also no relationship between publication year and the proportion of datasets with data problems that prevented reanalysis, or with the proportion of reproducible results.

We could not attempt reanalysis for 15 of the 86 eligible data sets because of obvious problems in the data file. These problems included the wrong data file being provided, missing data (individuals or variables), differences in the labels of variables between data files and published work, or unspecified subsetting of the data files prior to the analytical steps. While some of these problems could be solved through further communication with the authors, we wanted our study to reflect the long-term reusability of the data, when contacting the authors becomes increasingly difficult (Vines et al., 2014). Digital information is rapidly moving towards a more centralised online system (“the cloud,” Armbrust et al., 2010). Similarly, the responsibility for data preservation is being lifted from scientists to online repositories (e.g., Dryad (www.datadryad.org), figshare (www.figshare.com), NCBI (www.ncbi.nlm.nih.gov)). Given this paradigm shift, we recommend more attention be given to metadata quality and to the curation of the archived files (Michener et al., 1997). For instance, if data are size-adjusted or manipulated in other ways, both pre- and post-transformation data should be archived. Perhaps the most critical piece of information is the link between column labels in the data file and the variables described in the paper. We were unable to determine the correct columns or rows for 7% of datasets. While we were able to convert all data files to text format, the loss of metadata may stem from this conversion (in one case, this had to be typed by hand, because data file provided was from a scanned hardcopy of the data in a MSc thesis appendix). In line with previous authors on this topic (Borer et al., 2009; Whitlock, 2011), we recommend storing data in text-based data formats (such as comma- or tab-separated files), as these are non-proprietary and accessible across the range of statistical software packages. Also in line with previous papers, we recommend publishing the code used in the analyses as it is often difficult to provide a full description of the parameters used for a given analysis in the methods section of a journal article (Wolkovich, Regetz & O’Connor, 2012). This can be done as part of the Supplemental Information 1 or in online repositories such as GitHub (see Ram, 2013) or Zenodo, which provides a DOI and archives a permanent version of the code.

Among the 71 data sets that were suitable to be reanalysed, we were able to get within 1% of the published value for at least one of the three statistics that we focused on (PVE, PAC and the largest (absolute) coefficient) for 59 studies (83%). There were strong positive correlations between published and reanalysed values for statistics reported in DFA, which suggests that replication, in the broad sense, is possible when the proper metadata are provided and with adequate curation of the data file. Slight discrepancies could be due to differences in rounding, or differences in how data are handled by the various software packages. Although data file problems that preclude reanalysis appear to be associated with some software (particularly SAS, see Table 1), there was no effect of software on the reproduction of the published results in our reanalyses.

Evaluating whether the DFA metrics analysed here fall within 5% of the published values is, in our view, a reasonable test of reproducibility. However, it is uncertain how much the original conclusions from these studies would change based on the values we have obtained. The reproducibility of inference is an aspect of reproducibility that we admittedly did not explicitly address in this study. Additionally, while DFA was not always a central or essential component of the original study, its reproducibility is an important indicator of the underlying data’s quality and completeness. Such checks are a worthwhile consideration when archived data are being re-used for new purposes.

The reproducibility of the DFA varied depending on the summary statistic examined, ranging from 58% within 1% of the original value for the largest coefficient to 73% for the more complex PAC analyses, and 77% for PVE. The discriminant function coefficients were less likely to be reproduced, even when PVE and/or PAC matched. The procedures used to standardise model coefficients and calculate PAC differed among statistical packages and studies. For instance, if we had used only jackknifing for all PAC reanalyses, only 56% of published values would have been reproduced (results not shown). While this clearly does not invalidate the original results, it does highlight another obstacle to successfully reproducing the authors’ results: some summary statistics may be inherently harder to reproduce, particularly when there are numerous calculation methods, as is the case here, or when the estimation procedure makes use of stochastic numerical optimisation methods (e.g., Gilbert et al., 2012).

In comparison with our previous study of reproducibility of analysis using the genetic analysis program STRUCTURE (Gilbert et al., 2012), the proportion of studies with inadequate data or metadata was similar to the current reanalysis (17 of 60 studies (28%) in Gilbert et al., and 15 of 86 (17%) here). The proportion of studies where we could not reproduce the analysis was also broadly similar: 9 of 30 analyses (30%) in Gilbert et al., and 12 of 71 studies (17%) here. Despite the somewhat lower percentage of successful reanalyses, the correlation between published and reanalyzed results was consistently greater for DFA (r = 0.94–0.96) than for STRUCTURE (r = 0.59). In attempts to reanalyse microarray data sets, which are much more complex than morphological data sets, approximately half of the results could be reproduced from available data (Ioannidis et al., 2009). It is not surprising that analyses with more steps and parameter choices are harder to reproduce, and this is echoed within our study, where we had to explore a wide range of analysis options to obtain close matches for the most complex DFA statistic, PAC. At the same time, DFA represents a relatively simple, well documented scientific analysis, and it is likely that complex analyses with more subcomponents, larger datasets, complex software dependencies and a less objective decision process will require much more information (metadata) to re-analyse and eventually reproduce. The wider adoption of metadata standards like the Ecological Metadata Language (Michener et al., 1997, https://knb.ecoinformatics.org/#tools) would go a long way towards ensuring that crucial details about more complex datasets are not lost over time.

Shared data is an important substrate for science and is one of the levers that may be used to improve the reliability of research (Ioannidis, 2014). The system of having data re-users directly contact data generators to obtain access to their data has been in place for decades, and is absolutely necessary for data re-use within embargo periods (Roche et al., 2014), but it is not a long-term solution for the preservation of research data (Vines et al., 2014). We argue that in order for archived data to retain their full value, all of the necessary data and metadata must be stored at the time of archiving, which typically happens at or soon before/after publication. We have determined some of the common problems that can occur in self-archived data even when authors can be contacted and are able to share their data. The same factors are relevant to communal data archives. While sequence repositories such as NCBI Genbank have made the provision of metadata a key part of the submission, the decision of what additional information to archive to more generalised databases such as Dryad, figshare, Zenodo or GitHub, lies with the authors. The results presented here and those of previous studies (Savage & Vickers, 2009; Gilbert et al., 2012; Drew et al., 2013; Vines et al., 2013; Vines et al., 2014) illustrate the need for our research community to make data availability and curation a central part of the research and publication process.

Supplemental Information

Supplemental Information 1 Zip file containing example datasets, R code and readmes

Click here for additional data file.

We are extremely grateful to the authors who kindly provided their data, without which this research would not have been possible. We also thank our collaborators on the first part of this project, Florence Débarre, Michelle Franklin, Kim Gilbert and Jean-Sébastien Moore. We thank Michael Whitlock and Heather Piwowar for useful discussions during the planning of the project and Mary O’Connor for thoughtful comments on our manuscript.

Additional Information and Declarations

Competing Interests

Author Contributions

Tim Vines is the sole proprietor of the Molecular Ecology Editorial Office.

Rose L. Andrew conceived and designed the experiments, performed the experiments, analyzed the data, wrote the paper, prepared figures and/or tables, reviewed drafts of the paper.

Arianne Y.K. Albert conceived and designed the experiments, analyzed the data, reviewed drafts of the paper.

Sebastien Renaut conceived and designed the experiments, performed the experiments, analyzed the data, prepared figures and/or tables, reviewed drafts of the paper.

Diana J. Rennison and Dan G. Bock conceived and designed the experiments, performed the experiments, prepared figures and/or tables, reviewed drafts of the paper.

Tim Vines conceived and designed the experiments, performed the experiments, analyzed the data, wrote the paper, reviewed drafts of the paper.

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
