# Peer review of "Assessing the reproducibility of discriminant function analyses"

_PeerJ, doi:10.7717/peerj.1137_

## Round 0.1 · original submission · Major Revisions

While all reviewers were favorable in principle, they also had numerous comments. Reviewer 2 had the most extensive comments. I agree with that reviewer, in particular regarding the state of the code and the availability of the data/reproducibility of the work. I think that a paper on reproducibility of scientific works should be fully reproducible.

Reviewer 1 ·

Basic reporting

Is the article written in English using clear and unambiguous text and conform to professional standards of courtesy and expression?

Yes.

Does the article include sufficient introduction and background to demonstrate how the work fits into the broader field of knowledge? Relevant prior literature should be appropriately referenced?

Yes.

Does the structure of the submitted article should conform to one of the templates? Significant departures in structure should be made only if they significantly improve clarity or conform to a discipline-specific custom.

Yes.

Are the figures relevant to the content of the article, of sufficient resolution, and appropriately described and labeled?

Yes. Minor point: Figure 1 just illustrates the first row of Table 1, so should perhaps refer to Table 1 in the caption.

Is the submission ‘self-contained’?

Yes.

Experimental design

Does the submission must describe original primary research within the Scope of the journal?

Yes.

Does the submission clearly define the research question, which must be relevant and meaningful?

Yes.

Was the conducted rigorously and to a high technical standard?

Yes.

One weakness, acknowledged by the authors, is that we do not know what extent the conclusions of these papers would be affected by the discrepancies between the published and reanalyzed statistics. Since it would be quite difficult to make this judgement objectively and uniformly across the different studies, the authors' decision to leave this out is justifiable.

The thresholds of 1% and 5% are somewhat arbitrary, but the Figures allow the reader to gauge the distribution of differences between published and reanalyzed values. I was surprised (and reassured) to see that values far from the diagonal are actually quite rare.

Are the methods described with sufficient information to be reproducible by another investigator

Yes, and the authors are to be commended for including their R code (in Supp Mat).

Has the research been conducted in conformity with the prevailing ethical standards in the field?

Yes, as far as can be judged.

Validity of the findings

Are data robust, statistically sound, and controlled?

Yes.

Are the data on which the conclusions are based provided or made available in an acceptable discipline-specific repository?

The results for each study are provided in Table 2 (although it does not include the citations for a surprisingly large fraction of the authors who wished not to be identified). The underlying data that was reanalyzed for each study is *not* being made available; the authors should explain why and how access could be obtained.

Are conclusions appropriately stated, connected to the original question investigated, and limited to those supported by the results?

Yes.

Additional comments

This is a very well-executed study and well-written paper. It nicely complements and builds upon the authors' prior work showing the lack of availability of data for re-analysis and the repeatability of results from population genetic studies that use STRUCTURE. All the most relevant literature that I am aware of is cited.

Minor/typos:
l234. Add space before "("
l239. "Nature's Scientific Data" is not a generalized database but a journal. Do the authors intend figshare or zenodo?
Table 1 - I am still not sure I fully understand the meaning of the 3 columns under the heading "Reanalyzed".

·

Basic reporting

I think the paper is clear and well written. I make a few suggestions below that would in my opinion improve the clarity of the work, or point out minor errors.

In no particular order of importance:

1. Line 54 has an odd reference “Announcement: Reducing our irreproducibility”. Is that intentional?

2. Describe DFA
The paper is all about reproducibility, and uses DFA analyses as a test case (the reasons for this are clearly described). For me, the paper would have been clearer if the authors could describe in the introduction what DFA is, what it does, what it assumes, etc. Ideally, this would also come with an overview of, or a typical example of, the questions people try to answer using DFA. This overview could presumably be drawn straight from the studies from which datasets were used.

While I appreciate that the point of this paper is not to describe DFA per se, providing the naïve reader (like me) with an overview of the method would aid understanding throughout the text. Particularly when one reaches some of the more technical parts of the methods, which are hard for someone who is not well versed in DFA to judge.

3. Figure axes
A small point. The figures include a 1:1 line, which is useful for readers to visually judge the degree to which values could be replicated. However, because the figure axes do not span equal ranges, the 1:1 line is a little difficult to interpret visually. For example, the top right of Figure 3, and the bottom left of Figure 4, do not immediately look like 1:1 lines. This is easy to fix: just make the x and y axis limits equal in each plot (xlim and ylim in ggplot2). Secondary point: I do not know for sure, but it looks from the data like the values should not go below zero or above 100. If this is the case, then the 1:1 line probably shouldn’t go there either. Limits on the length of the line can also be set in ggplot2, independently of the axis limits.

4. Methods description
I think the description of the protocol could be a bit clearer if the authors stuck to ‘categorised’ rather than switched between ‘categorised’ and ‘assigned’. For example, by changing line 105 to read “we categorised the paper as ‘incorrect data file’”. Ditto for line 116.

5. Methods description 2
On Lines 161-162 it would help if the authors could refer directly to the classifications, rather than introduce related but not identical descriptions of the classifications.

6. On line 202 there are two phrases like this: “20 (80%) of cases”. The ‘of’ is out of place here. Perhaps clearer as e.g. “20 cases (80%)”.

6. Figure order
Figure 1 is not mentioned at all in the main text (though it is useful). Figures 2-4 are mentioned out of order as 2, 4, then 3 (i.e. the Figure numbers need changing).

7. Line 219: ‘(test P>0.2…’ The details of which test need to be provided.

8. Line 223. Please define what you mean by ‘entirely’. E.g. did every result match? To how many significant figures?

9. Encouraging?
I’m intrigued by the authors’ use of ‘Encouraging’. Perhaps I’m just a pessimist (I’m British), but that ~20% of studies had problems with datasets, and that this doesn’t seem to be improving, does not immediately seem encouraging to me. (Admittedly, it’s nice that it’s not getting worse).

10. Topic sentence line 241
The topic sentence for this paragraph had me expecting the paragraph to be about the 81% of datasets that were re-analysed. But the paragraph is instead about the problematic data sets. It might be clearer to start the sentence with e.g. “We rejected 19% of papers because of obvious problems with the data file”

11. Formats
Storing data in text based formats is fine. But not all text-based formats are equal. Can I suggest that the authors go further and recommend ‘human- and machine readable plain-text formats, such as .csv’

12. GitHub
GitHub is great, but it’s not a good place to archive published code. A GitHub repo can be changed or just deleted by the owner at any point. Thus, even if a paper includes a pointer to a specific commit, there is no guarantee that the repo will last. A better approach for publishing code is to archive the code used for the analyses (i.e. from a specific commit) using Zenodo. This archives the code properly, and provides a DOI. The particular version of the code used for analysis will then persist regardless of what the owner of the GitHub repo does.

13. Line 250: ‘figshare’ should be ‘figShare’.

14. Line 295: can you clarify in the text what you mean by ‘overly optimistic’.

15. Line 329: Nature’s Scientific Data is not a generalished database. It just publishes papers that have data stored in some other repo. Maybe just change this to figShare instead.

Experimental design

It seems like a very carefully considered study. I have one comment.

1. Excluding datasets
On lines 110-113 the authors state that they rejected datasets in which the variables were in a foregin language or ‘could not be matched to the variables reported in the paper’. This seems sensible. But they decided to accept studies with unlabelled columns, as long as there were the correct number of columns. This seems inconsistent to me. Specifically, columns with headers that don’t match the paper provide positive evidence that there’s an issue, so it makes sense to reject them. But papers without headers provide zero evidence on whether the correct columns are included. If anything, it seems that researchers who don’t label their columns are probably more likely to be those that don’t have great data hygiene, and may have issues with their data.

Can the authors clarify the reasons for this decision. Also, I’d like to see the numbers on how many studies had each of these problems (wrong headers, foreign language headers, no headers), and provide the information in the table.

Validity of the findings

This is, as it stands, impossible for me to assess (unless I’ve missed something here).

1. R code
The analyses in this paper are not, as far as I can tell, reproducible. The authors provide some R code which could be used to analyse one of their datasets. However, this R code seems to be written more for the authors than for readers. (E.g. comments like “## Load spreadsheet - see v1 to download from Google Drive”). The preliminary instructions are fairly inscrutable as well. Even if I had a dataset (which I don’t, see below) and could follow the preliminary instructions, I wouldn’t have any way of knowing if I’d done the same as the authorst. Thus, I have no way of reproducing what the authors did. This seems like a big oversight in a paper on reproducibility.

I would ask that the authors spend a bit of time cleaning up their comments. The code shouldn’t change, but as it stands it would be very hard to use even if I did have the data (see below). Some standard kinds of formatting would be useful: provide at the top of the script the information on who wrote the code, and detailed descriptions of the input(s) and outputs, ideally with a ‘quickstart’ section providing a commandline that should just work. This does not have to be onerous, but can be enormously helpful in enhancing reproducibility.

2. Data
Where is the data for this study? I am confused here. I have searched the paper, the PeerJ website, and the code. And I couldn’t find any mention of access to the data. Maybe I just missed it, if so, I apologies. But if so, I’d also ask that the authors make clear both in the ms and in the R code where one can get the data to reproduce the analyses.

If the data are not available, they should be made available. Specifically, the datasets from all of the studies need to be available in a suitable repository (i.e. with a DOI), as well as data on any decisions the authors made about how to analyse each of the datasets (i.e. the answers to the ‘preliminary’ questions at the top of the R code).

Additional comments

I like this paper a lot. It seems well thought out, and is clearly written.

My only major concern is that the the data is not available. Unless I have missed something here (in which case, sorry!), this renders a study on reproducibility completely irreproducible.

·

Basic reporting

All of my comments to the authors are in the attached PDF

Experimental design

Comments in attached PDF

Validity of the findings

Comments in attached PDF

Additional comments

Comments in attached PDF

---

## Round 0.2 · accepted · Accept

Thank you for your extensive revisions in response to the reviewer comments.

In preparing the final document for publication, please make sure to proof-read everything carefully once more. Two issues that I noticed:
- l. 157: "IAs"
- l. 278, reference "Ram 2013" does not appear in the bibliography.

·

Basic reporting

No further comments. Thanks for addressing all of my previous comments constructively.

Experimental design

No further comments. Thanks for addressing all of my previous comments constructively.

Validity of the findings

No further comments. Thanks for addressing all of my previous comments constructively.

Additional comments

Thanks for all of the additional effort. The R script produces a couple of minor errors when I run it, caused by some assumptions about file paths, but frankly I think this is expected when switching machines. To my mind the important thing is that the script is clear, and I can easily find and fix the errors. I do not think anything needs to be changed.